# Crisis Leadership: Political Leadership during the COVID-19 Pandemic

**Ataus Samad ***, **Khalil Al Jerjawi** and **Ann Dadich**

School of Business, Western Sydney University, Penrith South, NSW 2751, Australia
* Correspondence: a.samad@westernsydney.edu.au

**Abstract:** This article identifies leadership attributes that enable effective leaders to manage crises. Data were collected via semi-structured interviews with 13 Australian political leaders, including senators, members of federal and state parliament, premiers, ministers, and mayors of local governments. The findings suggest that, to be an effective leader during a crisis, political leaders need to be: visionary; courageous; calm; inspirational; ethical; empathetic; authentic; and resilient. Single leadership theories do not capture all the attributes necessary to lead during a crisis, suggesting the importance of different, complementary theories. The findings clarify what it takes for politicians to lead during a global crisis, like COVID-19. Furthermore, they provide a foundation to enable constituents to gauge their political leaders' leadership capacities. Despite extensive research on what it takes to lead, little is known about political leadership during a crisis. The study unveils the key attributes that are essential for political leaders to navigate a crisis, like the COVID-19 pandemic.

**Keywords:** leadership; leadership attributes; crisis leadership; political leadership; COVID-19





## 1. Introduction

Leadership is often tested during a crisis. It is thus appropriate to ask, what makes for effective political leadership during a crisis? Today, 'crisis' is often used interchangeably for situations such as disaster, accident, emergency, and myriad business problems, such as bankruptcy [1]. A crisis is generally defined as an unexpected event that is highly salient and disruptive [2–4]. Pearson and Clair defined crisis as 'a low probability, high-impact event that threatens the validity of the organization and is characterized by ambiguity of cause, effect, and means of resolution, as well as by a belief that decisions must be made swiftly' [3]. A leader frequently navigates crises. Political leaders represent their constituencies, making decisions at local, national, and/or international levels that significantly impact citizens' lives. The COVID-19 pandemic created a greater focus on leadership, increasing the burden on leaders to make decisions for the public in an uncertain time. Indeed, the COVID-19 pandemic represents a crisis [4].

During a crisis, people look for the best in their leaders. For example, while public perceptions of leadership during COVID-19 trended positively compared to perceptions before COVID-19 [5–7], Australians' trust in their government has declined since 2021. According to the Australian Institute of Management, people want to follow real people, and it is important for leaders to develop relational skills to manage complex situations [8]. It is therefore pertinent to examine leadership during this crisis, particularly that among politicians, given their role and responsibilities to their constituents.

This study clarifies how Australian politicians led during the COVID-19 pandemic. The rationale for this focus is threefold. First, there is limited scholarship on political leadership, particularly during a crisis. Second, politically-led organizations can provide a valuable understanding of generic processes in organizational behavior [9]. Third, political organizations 'expose fundamental problems connected with rationality and action and can teach us a great deal about fundamental problems and solutions in organizations' [10].

While there are several measures of leadership styles (e.g., transformational, transactional, authentic, ethical and servant leadership), there is no known measure of crisis leadership. Furthermore, recognized leadership styles have been criticized [11–15] and despite overlapping attributes among the leadership styles [16,17], it might not be feasible to measure leadership effectiveness during a crisis. It is therefore important to identify the attributes required for effective political leadership during a crisis. As such, this study clarifies the attributes of an effective leader in a crisis in Australia.

This exploratory study builds on the concept of crisis leadership while addressing the limitations of contemporary leadership theories in defining the attributes of effective leadership in crisis. The study was conducted during the COVID-19 pandemic. It involved interviews with 13 Australian political leaders at the local, state, and federal levels across the political spectrum. The study found that, during a crisis, leaders need attributes that might not align with a single leadership theory. This has research and practical implications. This article commences with a review of extant literature. It then presents the findings from this study and culminates with a discussion of the associated implications.

## 2. Literature Review

Historically, leadership is an omnibus term. It was applied to diverse roles, such as business executives, playground leaders, club presidents, and politicians, among others [18]. According to some, a 'leader occupies a particularly important position in some predefined institutional structure', while others proposed, 'the leader provides a common focal point around which the group may coordinate' [19]. Hogan and Kaiser argued that leadership generally denotes the people in charge [20], while Uhl-Bien and colleagues stated that leadership is dynamic and leaders are 'individuals who act in ways that influence this dynamic and the outcomes' [21]. This recognizes the important role of followers who provide leaders with legitimacy and power [19,22,23]. Leadership is salient when followers believe that others pay attention to the same leader. People turn to leaders for direction and the leader coordinates action.

Many theories have been proposed and debated to understand leadership [11–15,24]. These include the full range of leadership theory [24], servant leadership [25], authentic leadership [26], and distributive leadership [27,28]. But many of these demonstrate considerable overlap or limitations. For example, there are overlaps between authentic, ethical, and authentic transformational leadership [11,26,29–31]. Brown and Trevino [11] echoed this sentiment, demonstrating how particular leadership traits are found in ethical leadership, authentic leadership, spiritual leadership, and transformational leadership. They called for more research to assess whether other transactional elements of ethical leadership are important. There is also overlap between measures of ethical leadership and authentic leadership [16,17]. Additionally, 'authentic leadership is a root construct that can incorporate transformational and ethical leadership' [29]. Both transformational and authentic leadership contain an ethical component [32]. Alvesson and Einola studied authentic leadership as a 'problematic example of positive leadership' and identified fundamental flaws in this leadership theory [33]. Similarly, Uhl-Bien and colleagues argued that theories of the past century are 'largely grounded in a bureaucratic framework more appropriate for the industrial era' [21]. There are also instances where the same leadership style was measured differently [16,34]. As such, effective leadership is unlikely to be captured by one theory. An understanding of the attributes of an effective leader in a crisis would add value to contemporary leadership literature.

As political leadership is shaped by myriad interdependencies between leaders and followers, among leaders, within the organization, and in context, political leadership cannot be entirely explained with the trait, contingency, or situational theories of the past. Recent literature mentions crisis leadership, which focuses on 'the processes of how a crisis influences leaders, how leaders exert influence on the effect, cognitions, and behaviors of different stakeholders around times of crisis, and why some leaders are more effective than

others in crisis context' [4]. Therefore, it is pertinent to reexamine leadership in the current context of the COVID-19 pandemic.

While there is extensive research on leadership in different disciplines, political leadership is understudied [35,36]. Mintzberg [37] and Pfeffer [38] labelled organizations as political arenas and in the management literature, politics is viewed differently in different organizational contexts [10,36–42]. For example, Thompson and colleagues [42] discussed political skills in business organizations. Vigoda-Gadot discussed organizational politics, noting the effects of transformational and transactional leadership on employee performance [43]. While Bolmen and Deal talked of the political frame while discussing organizational leadership, Morrell and Hartley focused on a model of political leadership and argued that political leadership is understudied in the literature and therefore warrants attention [36,39]. Dion termed political leadership as 'most elusive' [40].

Despite some similarities between political leadership and leadership in non-political contexts, political leadership is different [36]. This is because politicians hold one of the highest levels of leadership in society as the people's representatives. Political leaders are usually democratically, elected to represent their communities and are thus vulnerable to deselection. They operate within a constitutional and legal framework, and their source of authority is the people's permission to govern [35]. They lead their community or nation through governance and policy formulation, navigating difficult local and international environments. Political leaders operate under different structures of scrutiny and accountability compared to other leaders in non-political contexts. The relationship between political leaders and their stakeholders is complex. Since they are in power through election, they require approval from those they govern and are responsible for serving their constituents. Hence, political leaders can be concerned with popularity, especially at election time. Furthermore, they are accountable to their political party, their opponents, the media, and interest groups, among others. It is further argued that the networks within which political leaders operate have their own regulations or norms; yet the environment of political leadership is often fluid [36]. Besides their constitutional roles of public service, as well as regulatory and enforcement roles, political leaders are subject to various challenges, like gaining consensus within their parties and the people they represent [44]. Political leaders juggle their need for consolidating their positions and maintaining a strong performance rating while introducing reforms. For example, Dion argued that, due to an increased demand for control over the leadership among party members in a democracy, the aforesaid challenges become more visible [40]. Hence, the findings of the study presented in this article will elucidate the attributes of effective leadership under crisis with a dedicated focus on political leadership. Following this review of leadership literature, the following section describes the study and the associated findings.

## 3. Methods

Given the aforesaid limitations of contemporary leadership theories and given this study was conducted during an unprecedented crisis, the study did not focus on specific leadership theories. Rather, it was exploratory, informing a larger study on crisis leadership, given the COVID-19 pandemic represents a crisis [4]. We applied a grounded theory approach to identify the leadership attributes essential for effective leadership in crisis. This was achieved with reference to the following interrelated open-ended questions, which were asked of all participants: what makes people effective leaders under challenging circumstances; what makes people bad leaders; and why do good leaders sometimes fail?

As self-selecting participants, the participants represented a convenience sample, based on their availability. Following clearance from the relevant ethics committee (reference number: H14128), potential participants were contacted via publicly available contact details and provided with an information sheet and a consent form. Thirteen political leaders (10 male and 3 female politicians) across the Australian political spectrum participated in this study. They included senators, members of federal and state parliament, premiers, ministers, and mayors of local governments. It has been recommended that qualitative

studies require a minimum sample size of at least 12 to reach data saturation [45,46]. Therefore, 13 interviews were deemed sufficient for the qualitative analysis and scale of this research. Furthermore, data were collected until thematic saturation was reached, whereby no new themes were apparent [47,48].

Because of the COVID-19 pandemic, interviews were conducted via web-conference at a mutually agreed time, following informed consent. Given the exploratory nature of this study, questions were open-ended, enabling us to investigate crisis leadership [49].

The recorded interviews were transcribed and deidentified. Using NVivo 12 Pro to manage the dataset, the transcripts were analyzed thematically using template analysis to identify relationships among the themes [50]. Responses were analyzed within a framework of constructionism, whereby understandings of the crisis leadership were understood to be socially constructed. This approach helped to give voice to the politicians' accounts while accounting for our reflective influence on data interpretation. Each transcript was analyzed systematically and compared with other transcripts to facilitate generalization.

Throughout this process, we primarily explored and determined coherence within the dataset and the extent to which the constructed themes deepened our understanding of leadership attributes during a crisis. Specifically, the transcripts were deconstructed into key quotes that exemplified experiences with and perceptions of crisis leadership. Guided by King, a codebook was established based on a literature review, the study focus, and the interview schedule [51–53]. We reviewed the transcripts and compared them to the initial codebook to identify convergent and divergent themes [52]. When discrepancies arose, we discussed and reconciled these. Latent coding was conducted to go beyond the descriptive level of analysis and identify underlying assumptions, ideas, and ideologies that shaped or informed the descriptive content. The quotes were grouped into codes, the codes were grouped into categories, the codes were compared, and the central themes were constructed to convey different perceptions and understandings of crisis leadership [54]. The original and new themes were incorporated into a final coding template [53]. Applying this method, the analysis became much more interpretive, requiring us to become creative. Content analysis was then conducted by highlighting the frequency of occurrence of a particular theme. The content analysis allowed us to verify theme saliency and variation.

Finally, to ensure the reliability of the findings, several approaches were used [55]. First, we invited academic colleagues with expertise in qualitative research to audit the empirical process. Second, expert peers were invited to review the interview questions, a random sample of transcriptions, and the coding structure. Third, peer researchers were also invited to crosscheck the analysis, ask critical questions, and review the patterns. These steps were taken to ensure the credibility of the conclusions made. We also conducted follow-up discussions with the participants to confirm themes, interpretations, and conclusions.

## 4. Results

The politicians collectively spoke of several attributes that aid effective leadership during a crisis. The word frequency of the attributes and sub-attributes that appeared from this research is presented in Figure 1.

The attributes and sub-attributes constructed from the interviews are summarized in Table 1. Table 1 also presents similarities with attributes and sub-attributes highlighted in other leadership theories. However, the table only highlights the attributes included in leadership measurement instruments that are similar to the attributes identified in this study. To maintain participant anonymity, excerpts are accompanied by a code (e.g., R-1, R-2, R-3).

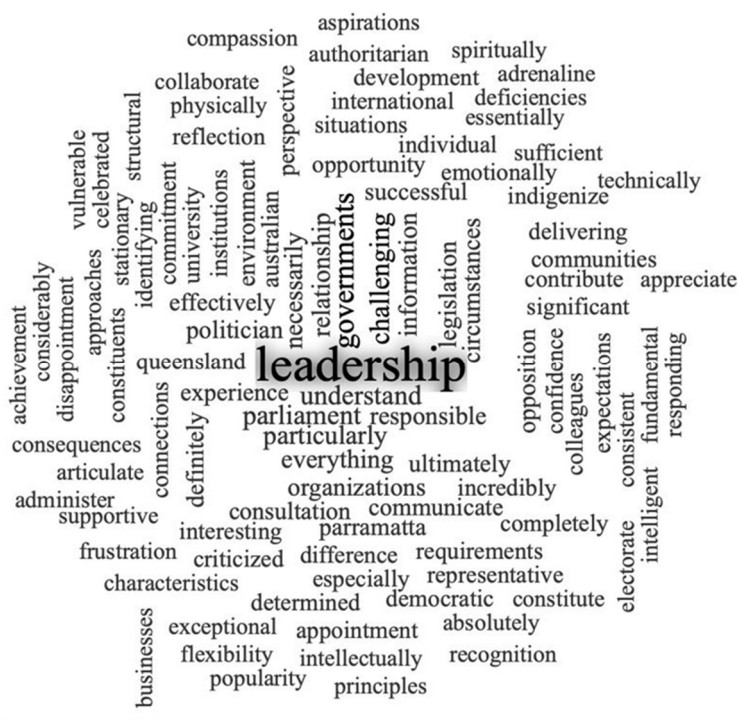

**Figure 1.** Word frequency of the attributes and sub-attributes associated with leadership.

**Table 1.** Key attributes of leadership in crisis compared to attributes identified in other leadership theories.

| Attributes | Sub-Attributes | Transformational Leadership [56] | Authentic Leadership [16,57] | Ethical Leadership [17,30] | Servant Leadership [34] |
|---|---|:---:|:---:|:---:|:---:|
| **Visionary** | | ● | | | ● |
| | Communicate | ● | ● | ● | |
| | Enthusiastic | ● | | | |
| | Promote trust | | ● | ● | |
| | Take people through the journey | ● | | | |
| | Influential | ● | | | |
| | Listen | | ● | ● | |
| | Persuasive | | | | |
| | Credible | | | | |
| **Courageous** | | | | | ● |
| | Prevent fear from leading to paralysis through analysis | | | | |
| | Maintain firmness in pursuing and attaining goals | | | | |
| | Truthful despite criticism | | | | |
| | Adhere to values and principles | ● | | | |
| | Determined | | | | |
| **Calm** | | | | | |
| | Emotional intelligence | | | | |
| | Optimistic | ● | | | |
| | Understand, analyse, and reflect | ● | ● | | |
| | Avoid impulsive and rushed decisions | | ● | | |
| | Self-control | ● | | | |
| | Avoid being overwhelmed by anger and nervousness | | | | |

**Table 1.** *Cont.*

| Attributes | Sub-Attributes | Transformational Leadership [56] | Authentic Leadership [16,57] | Ethical Leadership [17,30] | Servant Leadership [34] |
|---|---|:---:|:---:|:---:|:---:|
| **Inspirational** | | ● | | | |
| | Connect with people's emotions | ● | | | |
| | Self-awareness | | | | |
| | Understanding peoples' feelings | | | | |
| | Attracts people's respect | | | | |
| | Value-driven | ● | ● | ● | |
| | Responsible | | | ● | |
| | Promote sense of purpose | | | | |
| | Engage with followers | | | | |
| **Ethical** | | ● | | ● | ● |
| | Trustworthy | | ● | ● | |
| | Stimulate confidence | ● | | | ● |
| | Honest | | | | |
| | Respectful | | | | |
| | Empowering people | | | | |
| | Integrity | | ● | | |
| | Lead with fairness | | | | |
| | Pursue justice | | | | |
| **Empathetic** | | ● | ● | | |
| | Humble | | | | ● |
| | Respectful | | | | |
| | Understanding | | | | |
| | Emotional atonement | | | | ● |
| | Thoughtful | ● | | | |
| | Considerate | | | | |
| | Appreciative | ● | | | |
| **Authentic** | | | ● | ● | ● |
| | Self-actualised | | | | |
| | Genuine | | | | |
| | Firm and precise decision-maker | | | | |
| | Lead with heart and integrity | | | ● | |
| | Pursue purpose | | | | |
| | Perceive and accept criticism | | ● | ● | ● |
| | Balancing rational reasons and empathetic emotions | | | | |
| | Employ strengths and understand limitations | | | | |
| | Provide insights for future leaders | | | | |
| **Resilient** | | | ● | | ● |
| | Agile | | | | |
| | Develop a sense of purpose | | | | |
| | Facing obstacles | | | | |
| | Ability to bounce back | | | | |
| | Learn from failures | | | | |
| | Perceive challenges as learning opportunities, not obstacles | | | | |

## 4.1. Visionary

Taking the people through a journey means a leader must set the vision and communicate it to team members. An effective leader will define the vision and communicate it to foster commitment and enthusiasm within the community during a crisis. Issues associated with vision are shown in Figure 2.

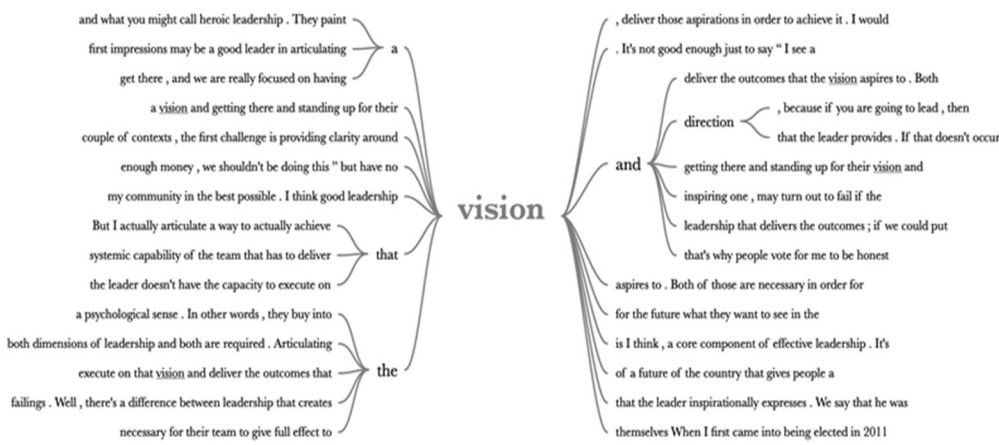

**Figure 2.** Word tree associated with vision.

A good leader sets the goal in the followers' minds and communicates to inspire and build trust. Therefore, effective leadership during a crisis requires one to take people through a journey, guiding them in every step and demonstrating what is needed to achieve the goal:

> *The first challenge is providing clarity around vision and direction, because if you are going to lead, then you need to lead towards goals that have to be clear in leaders' minds as well as perceived clearly by their followers (R-3).*

> *You've got to be prepared to take people through that journey and talk to them and understand them (R-1).*

These leaders use their influence to inspire their team to get things done. Successful leaders are visionary and use their influence to make people move towards a specific goal. A leader should have the ability to listen to people and effectively communicate their vision for taking people along a journey:

> *to be a good leader is to be able to listen to what the community needs and wants, and then be able to communicate a roadmap (R-8).*

> *When being listened to, people become more willing to follow and commit to your leadership, as they feel valued and appreciated (R-11).*

When politicians fail to listen and take people on the journey, no matter how competent they are, they consequently fail to achieve the proposed vision. This can have implications for how others perceive their leadership style and their ability to continue their political role:

> *What on first impressions may be a good leader in articulating a vision and inspiring one, may turn out to fail if the leader is not an active listener and doesn't have the capacity to persuade people to walk through the journey . . . the execution of the vision and the delivery of the outcomes will be at risk and rarely to be achieved (R-3).*

### 4.2. Courageous

Leadership in crisis presents a lot of uncertainty, requiring one to put aside fear and manage the situation. Being courageous means remaining firm in pursuing what is right and being willing to speak the truth, despite criticism. Leaders who stick to accepted values and norms emerge as saviors in adverse situations. By prioritizing followers' wellbeing and empowerment, leaders emerge as change agents because they are determined to tackle the situation:

> *It takes more courage in difficult times to stick to what you have committed to achieving because you are going to be subjected to a lot of criticism (R-1).*

> *I think you need to stand up for righteousness . . . normally we would like to take good decisions with available information. Sometimes, that information changes over time*

> *or the situation changes ... You'd never just stick to a decision just because it is an ideological view (R-5).*

Rather than stick to an ideological view due to fear of criticism, courageous leaders are flexible and make decisions they believe are in the best interest of the community. The word tree associated with courage is shown in Figure 3.

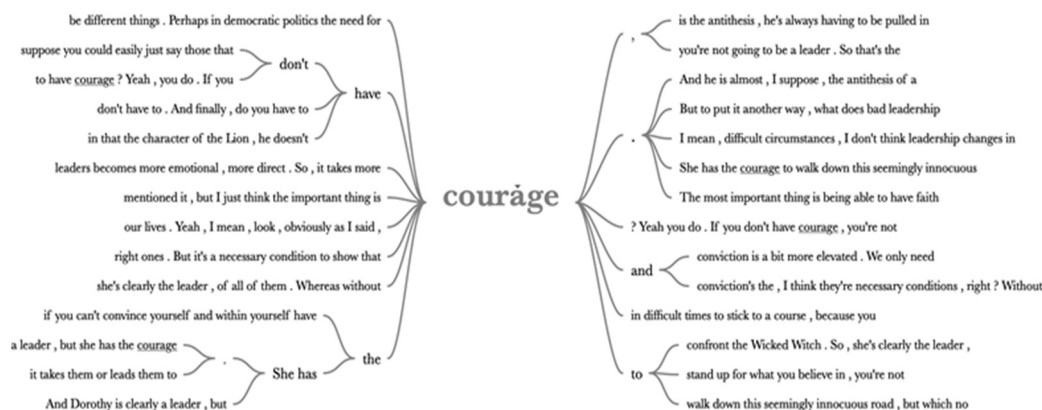

**Figure 3.** Word tree associated with courage.

### 4.3. Calm

Remaining calm during a crisis means that a leader should practice self-awareness and emotional intelligence and be optimistic about the future. By demonstrating awareness, a leader understands the situation, analyses it, and reflects on the decision that would be most effective. Rather than being impulsive and rushing into action, a leader must remain composed and understand the challenge. Furthermore, emotional intelligence allows leaders to practice self-control and avoid being overwhelmed by anger. Leaders will likely make irrational decisions whenever they become nervous:

> *No matter what happens, you got to stay calm. Things can be going against you, but you can't let that sort of taking over. And people outside can see you getting nervous and losing temper. You've just got to stay calm, maintain self-management, and never panic and make rash decisions (R-7).*

When in a stressful scenario, it is essential to pause and assess the crisis while engaging the 'rational mind' before responding. Being optimistic in a crisis provides hope that, regardless of the difficulty, the team will emerge victorious. Future leaders can use these findings by exercising self-awareness and emotional intelligence when faced with a crisis, as this will create calmness and a better approach to the issue. Words that are associated with calm as revealed through the analysis are shown in Figure 4.

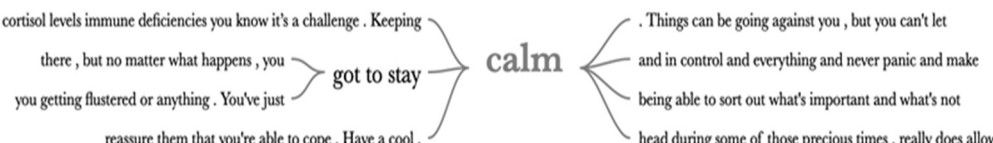

**Figure 4.** Word tree associated with calm.

### 4.4. Inspirational

Inspirational leaders can attract followers' respect because they connect with their emotions while clearly communicating a goal. They have self-awareness and understand others' feelings, and this can be particularly important during difficult times. Furthermore, an inspirational leader will attract subordinates' respect because they listen, care, and motivate. Being inspirational also means that a leader remains value-driven and has a responsibility and a deep sense of purpose to implement change:

*[Former Australian Prime Minister] Kevin Rudd was very, popular in the community, but not so popular with the people who worked with him as he failed to inspire them. Compared to someone like Bob Hawke [a former Australian Prime Minister], who was wildly popular in the community and also popular amongst his peers as an inspirational leader, further to validating the impression to people that he was one of them (R-11).*

*Inspirational leaders earn people confidence and respect, and people can relate to them by expressing that, 'We respect and have confidence in this person because of what they do and aspire to do' (R-10).*

The goal is to engage followers, listen to their needs, and inspire them toward the vision. The ability to genuinely listen and interact with followers enables them to stand out as reputable. For example, an inspirational person will mentor others, develop them, and build trust.

### 4.5. Ethical

Ethical leadership practices are essential to building trust and confidence among followers. This means the team members look to the leader as a role model – this reinforces the need to remain trustworthy and honest. Ethics allows the leader to establish a culture of respect and trust, enabling the people to have a positive reputation. Therefore, winning followers' trust and respect requires strong integrity, fairness, and justice before inspiring the team to behave ethically:

*There are a number of politicians who are in jail because of unethical behavior; that is evident and a glaring example and lesson (R-2).*

Conversely, unethical behaviors, such as integrity issues, can make people lose confidence in a leader. When political leaders distinguish themselves from their responsibilities as leaders, ethical political leaders are empowered to make decisions as leaders, free from the temptation of multiple identities, justifications, and the confusion of actions with intentions. Conversely, ethical individuals who fail to make this distinction, run the risk of making the wrong decisions for the wrong reasons [58].

### 4.6. Empathetic

According to the participants, being empathetic means being humble and respectful, while also taking others' views. By showing respect and understanding others' emotions, a leader creates a powerful bond that enables followers to embrace a goal. When a leader identifies with others' views and involves them in key decisions, they feel valued and respected:

*I've seen that the most successful leaders are the ones who embrace the views of others, who are consultative, who are empathetic, who [are] . . . decisive, but be inclusive of a team, and therefore someone the team can look up to and say they represent us (R-11).*

*I do not think just because you are 'the boss' that it gives you that unquestioned power to exercise. I think the respect of your peers and the people that vote for you is a far more legitimate source of power (R-5).*

Empathy offers a sense of emotional atonement, allowing people to express their feelings and communicate about a situation. Whenever there is a crisis, it is better to invite the team to share their emotions and listen to them. These outcomes imply that leaders must be sensitive to followers' emotions, showing them respect and encouraging them to express their views.

In political leadership, empathy is essential for connection and leading with compassion. Yet respondents suggested that political leaders would find it difficult to make decisions if they over-relied on empathy and prioritized others' emotions. Too much empathy can make it difficult to consider the greater good, as political leaders often understood empathy as echoing others' emotions. Empathy in political leadership might provoke

bias and compromise rational decisions. However, the participants did not clarify their interpretation of too much empathy.

### *4.7. Authentic*

Understanding one's strengths and weaknesses and being self-actualized are essential characteristics of an authentic leader. When faced with difficult situations, a leader must stand up and make genuine and firm decisions that promote others' best interests. Such moves require one to lead with the heart, have integrity, and pursue purpose and values based on self-awareness. They strive to be real and make positive decisions, even if they receive criticism:

> *I think the biggest challenge of leadership is having to make the hard decisions, knowing that there are times when you've got to make some decisions that will have a partial negative impact on people and that's a hard one, especially if you lead with your heart (R-11).*

> *Being a leader almost invariably involves you being criticized, because you're doing things in a different way (R-9).*

It is not easy for a leader to please everyone. Hence, being authentic requires one to balance reason with emotions and make genuine decisions that positively impact followers. Authenticity requires leaders to understand their strengths and limitations.

For effective political leadership, the manifestation of the authentic self is crucial. However, authenticity as a concept has often been misread and misinterpreted, not least by political leaders themselves. Political leaders often perceive authenticity as a natural trait, whereby a person is either authentic or not. Conversely, authenticity is a trait that others must attribute to you. Authenticity often puts political leaders in a position where they should be true to themselves and others. However, the idea of authentic political leader undermines the limitations and inadequacies of political leaders as human beings. It emphasizes on the idea collective self rather than one's true self which may impede subjectivity to both political leaders and followers [59].

### *4.8. Resilient*

Leaders experience uncertainty and hardship during a crisis, requiring them to be agile, flexible, and develop a sense of purpose. To be resilient means standing firm and quickly recovering from challenges. Sometimes, leaders can make decisions that are erroneous, resulting in criticism. In their daily experiences, they face obstacles, but are required to bounce back and drive their team towards a goal:

> *You got to be resilient. You got to be able to get knocked down and then get back up again. It's like a boxer; a boxer will get knocked down and has to get back up again to keep on going. And it's also about, never stop learning . . . the biggest one of all the basic things is to be agile (R-1).*

In each problematic situation, a leader must learn from failure and use the lessons acquired to navigate challenges. Resilience is commonly discussed as an attribute that individuals either own or do not – this in turn awards primacy to the individual, rather than the influential role of context. Here, the internal dynamics of the political system itself and the depolarized nature of resilience are overlooked [60,61]. Consequently, political leaders are often blamed for factors that are shaped by political, economic, and social forces.

### 5. Discussion

The significance of leadership during a crisis, like the COVID-19 pandemic, is more important than ever. The pandemic is a unique situation that disrupted normality, encouraging people to look for purpose and direction. This represented a challenge for political leaders around the world. This unique situation was appropriately considered a crisis by Wu and colleagues in their research on crisis leadership. This study examined leadership

through the eyes of Australian politicians to identify the attributes they considered to be essential for leadership during a crisis [4].

Leaders can experience a crisis when a situation is volatile, uncertain, complex, and ambiguous [62]. For example, during a war, natural disaster, or pandemic, leaders might not have the appropriate solutions to manage the situation; yet the population looks to their leaders to lead them out of the crisis. Although populations and political parties seldom consider crisis leadership skills when choosing their leader, they rely on their leader more than ever when a crisis erupts. This reinforces the importance of this study.

This study, which involved Australian politicians at different levels of government, revealed the attributes essential for crisis leadership. Reflecting extant literature, these included: being able to take people along a journey using vision and effective communication [16,24,27,57]; being inspirational and motivating [24,34,63,64]; authenticity and trustworthiness [16,17,65]; resilience and flexibility [16,34]; empathy [24,57]; being ethical [11,17,24]; courage [34,63,66]; and calmness [67]. Although these attributes are recognized by established leadership theories, this study brings them together to clarify those that are essential during crisis in a political context. For example, this study found that attributes such as being visionary, courageous, calm, and inspirational are the most important attributes in crisis, while being ethical, empathetic, authentic, and resilient are also important, but to a lesser degree.

In the context of crisis leadership, Haslam and colleagues argued that a sense of togetherness or taking people on a journey is essential for leaders to navigate a crisis [68]. Classical research on leadership emphasized different issues, such as the achievement of specific goals and leader-follower interaction [69–71]. Likewise, distributive leadership emphasized vision and evaluation [28]. Although the full range of leadership theory recognized inspirational motivation, cognitive reward, individualized consideration, and individualized influence as elements of transformational leadership, it did not include empathy, courage, or flexibility [24]; yet these are important for crisis leadership. Emel argued that a leader must remain true to personal values, be optimistic about the future, and maintain self-confidence [66]. During a crisis, people need support and compassion and a leader who cares for their feelings and respects them [72]. In a study on the relationship between character and political effectiveness, Sergent and Stajkovic found female governors in the United States of America cultivated more empathy and confidence through their COVID-19 briefings to effectively lead their states during the pandemic [73]. Exhibiting empathy, compassion, concern, and care for citizens' welfare enable political leaders to 'forge deeper connections with their constituents' [74]. Again, authentic leadership theory [27] focused on transparency, listening, and ethics, which was also indicated in this study. Servant leadership focused on motivation, freedom of action or flexibility, risk-taking or courage, care for employees' minds, and ethical behavior [30,34]. These too reflect the findings from this study. Similarly, Chipchase and Miller emphasized remaining calm in crisis [67]. Crisis leadership might therefore need a different measurement tool.

The findings from this study suggest that attributes that are recognized by different leadership theories are likely to be required to be an effective political leader during a crisis. This is because single leadership theories have a limited capacity to account for all the attributes identified in this study. The findings also pave the way for a crisis leadership measurement tool to enable scholars, practitioners, and policymakers to measure leadership effectiveness in a crisis.

Political leadership has its unique characteristics and challenges. As elected representatives of their community, political leaders lead their constituents during peace and crisis, and they are particularly relied on during the latter. It is therefore important to have a clear understanding of what constitutes effective crisis leadership. While context matters, so too does a leader's attributes [36].

### 5.1. Limitations

A limitation of this study is its reliance on political leaders' self-perception of leadership during a crisis. Thus, capturing constituents' perceptions represents an opportunity for future research. Furthermore, given the reliance on qualitative data, the findings are likely to be limited. To validate and test the reliability of the attributes and their sub-attributes, further study is needed using mixed methods with a larger, representative sample.

### 5.2. Practical Implications

Notwithstanding the limitations of this study, the findings presented in this article have (at least) two key practical implications, particularly for politicians. First, given the importance of the identified attributes, the findings suggest there might be value in professional development opportunities that bolster these. Given their demonstrated benefits, these might include coaching [75], reflective practice [76], emotional intelligence training [77], and communities of practice [78].

Second, politicians and their parties might find value in critically reviewing and potentially revising their protocols to ensure the context in which they work is conducive to the identified attributes. While the attributes reported in this article are largely at an individual level, it would be naïve to assume that the enactment of these attributes are the sole responsibility of individuals. Behaviors do not occur in a vacuum – as such, it would be prudent to critique accepted ways of working, lest they compromise politicians' ability to be visionary, courageous, calm, inspirational, ethical, empathetic, authentic, and resilient.

### 5.3. Conclusions

The COVID-19 pandemic exposed the need for effective leadership in all sectors, particularly in politics. Politicians at different levels of government have a significant role in leading their communities, particularly during a crisis. Crisis leadership is essential, particularly given the implications on public wellbeing. Despite extensive research on leadership, there is limited research on what it takes to be a political leader during a crisis. This study found that a single leadership theory is unlikely to capture all the attributes necessary to lead during a crisis – instead, the essential attributes are likely to be associated with different leadership theories. This suggests the need for complementary leadership theories to understand crisis leadership.

**Author Contributions:** Conceptualization, A.S.; methodology, A.S. and K.A.J.; software, K.A.J.; validation, A.S., K.A.J. and A.D.; formal analysis, A.S. and K.A.J.; investigation, A.S.; resources, A.S.; data curation, A.S.; writing—original draft preparation, A.S. and K.A.J.; writing—review and editing, A.S., K.A.J. and A.D.; visualization, A.S.; supervision, A.S.; project administration, A.S.; funding acquisition, A.S. All authors have read and agreed to the published version of the manuscript.

**Funding:** It was funded by Western Sydney University. The fund was from Western Sydney University DVC Research, hence mentioned as DVC Research Fund. The fund is specifically known as 'DVCR Initiative Grants'.

**Institutional Review Board Statement:** The study was conducted according to the guidelines of the Declaration of Helsinki, and approved by the Institutional Review Board Western Sydney University Human Research Ethics Committee (HREC) under the Australian National Health and Medical Research Council (NHMRC). HREC Approval Number: H14128, Initial approval 12 February 2021, extension granted on 19 May 2021.

**Informed Consent Statement:** Informed consent was obtained from all subjects involved in the study.

**Data Availability Statement:** The data presented in this study are available on request from the corresponding author, but subject to approval from the Western Sydney University Human Research Ethics Committee. The data are not publicly available due to ethical reasons of privacy and anonymity of the respondents since the topic is sensitive and disclosure of the data may reveal the identity of the respondents.

**Conflicts of Interest:** The authors declare no conflict of interest.

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
