# Peer review of "Crisis Leadership: Political Leadership during the COVID-19 Pandemic"

_sustainability, doi:10.3390/su15010266_

Round 1

Author Response

Dear Reviewer, 

We would like to offer our deepest appreciation and to sincerely thank you for offering us an opportunity to revise and resubmit our manuscript. We have carefully read through your excellent feedback, and the comprehensive feedback and we have now revised our manuscript. We hope that we have addressed all the concerns and that we have now built a stronger and compelling case for the importance and unique contribution of our study to meet the standards for publishing in your very esteemed journal. 

Copy of the rejoinder is forwarded for your consideration please. 

Best regards

Authors

Reviewer 2 Report

This study into Australian political leadership during crises is timely, the article is very readable and of good quality - it was a joy to review.

The authors' contribution, in my view, are twofold, first in their discussion aimed at future leaders (e.g. p7, lines 299-301; p8, lines 346-8), and second, by stating how their research adds new knowledge (e.g. p7, lines 285-6; p9, lines 396-404) to practice and theory.

RE study limitations: I wondered if some additional considerations regarding the study 'limitations', (see p9, sentence spanning lines 414-5), are to first explore 'constituents' impressions of actual effective political leadership during a crises' compared to 'how politician's self-reported their leadership', and second, to look at how both populations expressed 'ideal forms of poltical leadership during a crises'. 

Conclusion, page 10, following the last sentence, perhaps the authors might want to expand on the significance of their study .... as during these ongoing times of great uncertainty I thought this study also provided useful insights for future leaders at any level to consider.  

I'm sure it's been picked up already by other reviewers, but there are some strange word hyphenations e.g. 'percep-tions' page 1, line 39; other instances occur on p1, line 42; p2, lines 49, 50; p4, lines 161, 162, 169, 175, 186, 187; p9, lines 420, 422

delete extra line space, page 9, line 393

check line 609, citation 91

Author Response

Dear Reviewer, 

We would like to offer our deepest appreciation and to sincerely thank you for offering us an opportunity to revise and resubmit our manuscript.

We have carefully read through your excellent and comprehensive feedback  and we have now revised our manuscript. We hope that we have addressed all the concerns  and that we have now built a stronger and compelling case for the importance and unique contribution of our study to meet the standards for publishing in your very esteemed journal.

The attached rejoinder is forwarded for your consideration please. 

Best regards

Authors

Reviewer 3 Report

1.      Introduction: the authors need to state the problem statement, purpose of the research, brief findings, and contribution to the literature.

2.      Literature Review: the authors need to state exactly which theories the leadership attributes come from. In other words, the literature review should lay the theoretical foundation for the interview questions.  The readers do not know from which theories the specific interview questions are drawn from.

3.      Method: Why did the authors select 13 interviewees? There has to be theoretical explanation for this number. What are the selection criteria? What are the demographics of the politicians? What are the interview questions? Are they open-ended questions?

4.      Results: The authors need to give the answers to each interview questions, then found the most common answers to the interview questions. Since the interview questions are not given, the readers do not know which attributes are more significant than others. The authors need to make a list of all the leadership attributes and indicate which ones are “most important”, “important”, “less important”, and “unimportant”, etc.

5.      Discussion: the authors need to state the most important attributes and how they are compared with prior literature.

6.      Conclusion: The authors need to explain the significance of the research, the key findings, and the implications.  

Author Response

Dear Reviewer, 
We would like to offer our deepest appreciation and sincerely thank you for offering us an opportunity to revise and resubmit our manuscript.
We have carefully read through your excellent and comprehensive feedback and we have now revised our manuscript. We hope that we have addressed all the concerns and that we have now built a stronger and compelling case for the importance and unique contribution of our study to meet the standards for publishing in your very esteemed journal.
The attached rejoinder is forwarded for your consideration please. 
Best regards
Authors

Round 2

Reviewer 3 Report

This paper has been revised satisfactorily. 

Author Response

Point 1: The reviewer expressed that all observations in round 1 were addressed satisfactorily. However, against the question ‘Are all the cited references relevant to the research’, the reviewer indicated that the standard of referencing can be improved.

Response 1: Please provide your response for Point 1. (in red)

We checked all references in the text and in the reference list and removed any references that were either less important or were not cited appropriately. We also corrected references where applicable. References are cited in the text as advised.